# Microvesicles from quiescent and TGF-β1 stimulated hepatic stellate cells: Divergent impact on hepatic vascular injury

**Jianlong Xie**[1,2], **Zhirong Ye**[1], **Xiaobing Xu**[3], **Anzhi Chang**[1,3], **Ziyi Yang**[1,4], **Qin Wu**[2], **Qunwen Pan**[3], **Yan Wang**[3], **Yanyu Chen**[1,5], **Xiaotang Ma**[3☙], **Huilai Miao** ![ORCID][1,5,6☙] *

**1** Department of Hepatobiliary Surgery, The Second Affiliated Hospital of Guangdong Medical University, Zhanjiang, Guangdong, China, **2** Department of Cardiothoracic Surgery Center, Affiliated Hospital of Guangdong Medical University, Zhanjiang, Guangdong, China, **3** Guangdong Key Laboratory of Age-Related Cardiac and Cerebral Diseases, Institute of Neurology, Affiliated Hospital of Guangdong Medical University, Zhanjiang, Guangdong, China, **4** Department of Gastrointestinal Surgery, The Second Affiliated Hospital of Guangdong Medical University, Zhanjiang, Guangdong, China, **5** Key Laboratory of Liver Injury Diagnosis and Repair, Guangdong Medical University, Zhanjiang, Guangdong, China, **6** General Surgery, Liaobu Hospital, Dongguan, Guangdong, China

☙ These authors contributed equally to this work.
* miaohl-gdwk@gdmc.edu.cn

**Data Availability Statement:** All relevant data are within the manuscript and its Supporting Information files.

## Abstract

### Background

This study evaluated the effect of microvesicles(MVs) from quiescent and TGF-β1 stimulated hepatic stellate cells (HSC-MVs, TGF-β1HSC-MVs) on $H_2O_2$-induced human umbilical vein endothelial cells (HUVECs) injury and $CCl_4$-induced rat hepatic vascular injury.

### Methods

HUVECs were exposed to hydrogen peroxide ($H_2O_2$) to establish a model for vascular endothelial cell injury. HSC-MVs or TGF-β1HSC-MVs were co-cultured with $H_2O_2$-treated HUVECs, respectively. Indicators including cell survival rate, apoptosis rate, oxidative stress, migration, invasion, and angiogenesis were measured. Simultaneously, the expression of proteins such as PI3K, AKT, MEK1+MEK2, ERK1+ERK2, VEGF, eNOS, and CXCR4 was assessed, along with activated caspase-3. SD rats were intraperitoneally injected with $CCl_4$ twice a week for 10 weeks to induce liver injury models. HSC-MVs or TGF-β1HSC-MVs were injected into the tail vein of rats. Liver and hepatic vascular damage were also detected.

### Results

In $H_2O_2$-treated HUVECs, HSC-MVs increased cell viability, reduced cytotoxicity and apoptosis, improved oxidative stress, migration, and angiogenesis, and upregulated protein expression of PI3K, AKT, MEK1/2, ERK1/2, VEGF, eNOS, and CXCR4. Conversely, [TGF-β1]HSC-MVs exhibited opposite effects. $CCl_4$- induced rat hepatic injury model, HSC-MVs reduced the release of ALT and AST, hepatic inflammation, fatty deformation, and liver

**Funding:** National Natural Science Foundation of China general project (NO. 82070637). The funders had no role in study design, data collection and analysis, decision to publish, or preparation of the manuscript.

**Competing interests:** The authors have declared that no competing interests exist.

fibrosis. HSC-MVs also downregulated the protein expression of CD31 and CD34. Conversely, TGF-β1HSC-MVs demonstrated opposite effects.

## Conclusion

HSC-MVs demonstrated a protective effect on $H_2O_2$-treated HUVECs and $CCl_4$-induced rat hepatic injury, while TGF-β1HSC-MVs had an aggravating effect. The effects of MVs involve PI3K/AKT/VEGF, CXCR4, and MEK/ERK/eNOS pathways.

## Introduction

Severe, acute, and persistent chronic liver injury may lead to liver fibrosis, potentially culminating in liver cancer or failure [1]. The microenvironmental alterations during liver injury repair are complex, with sinusoidal endothelial cells and non-parenchymal cells contributing to liver fibrosis during the repair process [2]. In normal liver, HSCs maintain a non-proliferative, quiescent phenotype. Following liver injury or culturein vitro, HSCs become activated, trans-differentiating from vitamin-A-storing cells to myofibro-blasts, which are proliferative, contractile, inflammatory and chemotactic. They also synthesize and secrete large amounts of extracellular matrix (ECM). The excessive accumulation of ECM leads to liver structural remodeling and fibrosis formation [3].Studies have shown that Hepatic stellate cells and endothelial cells maintain hematopoietic stem cells during liver development through the production of stem cell factor (SCF) [4]. Moreover, HSCs secrete matrix metalloproteinases (MMPs), different MMPs and their inhibitors have been explored in preclinical studies for the treatment of liver diseases by degrading the most abundant fibrotic extracellular matrix (ECM) protein col-I, thereby facilitating liver damage repair and regeneration [5]. Recent studies, including those in this project, have demonstrated that HSCs release MVs and non-cellular components to regulate the functions of liver cells and vascular endothelial cells, highlighting the pivotal role of HSC-derived MVs as communication carriers in the regulation of hepatic cell injury repair, which has garnered significant research attention [6, 7].

MVs are a type of extracellular vesicle formed by budding and are currently recognized as new communication carriers between cells [8]. MVs contain various signaling molecules such as proteins and nucleic acids, released upon binding to recipient cells, playing crucial roles in regulating cell morphology and function [9]. The composition of MVs is determined by the state of the producing cells and closely correlates with their functions [10]. Our previous study demonstrated that MVs released by HSCs in their quiescent state (HSC-MVs) can inhibit caspase-3 protein expression and significantly reduce AST, ALT, and LDH levels, thereby preserving liver cell function in acetaminophen(APAP)/$H_2O_2$-injured liver cells [7]. However, MVs generated by TGF-β1 activated HSCs (TGF-β1HSC-MVs) led to a dose-dependent down-regulation of the PI3K/Akt and Erk1/2 pathways, along with the promotion of caspase-3 expression, ultimately leading to hepatocyte apoptosis when damaged hepatocytes were co-cultured with [TGF-β1]HSC-MVs [11]. These findings suggest that HSC-MVs can protect damaged hepatocytes, while TGF-β1HSC-MVs can promote liver cell injury.

HSCs, originally identified by von Kupffer in 1876, are localized in the subendothelial space of Disse, interposed between liver sinusoidal endothelial cells (LSECs) and hepatocytes; they represent ~10% of all resident liver cells [12]. There is a close relationship between LSECs and HSCs. LSECs release vascular endothelial growth factor (VEGF), inducing proliferation of HSCs and angiogenesis after liver parenchymal injury. VEGF promotes fiber formation and

may also be necessary for liver tissue repair and fibrosis resolution [3]. HSC-MVs protects against liver cell damage, while TGF- β 1HSC-MVs exacerbate liver cell damage. What is their impact on liver sinusoidal endothelial cells? Further clarification is still needed. This article uses $H_2O_2$ to induce HUVECs injury and $CCl_4$ to induce rat liver injury, further investigating the effects of HSC-MVs and TGF-β1HSC-MVs on injured endothelial cells and intrahepatic vessels.

## Materials and methods

### Cell lines and culture conditions

LX-2 cells (P4) were obtained from Jennio Biological Technology (JBT, Guangdong, China) and used for MVs preparation. HUVECs (P4) were sourced from the American Type Culture Collection (ATCC, Manassas, VA, USA). P5 to P10 cells were used for experiments.

LX-2 cells were cultured in Dulbecco's modified Eagle's medium (DMEM, Gibco, Cat#: 8120365), while HUVECs were cultured in an F-12K basal medium (Gibco, Cat#: 11330032) supplemented with 10% fetal bovine serum (FBS, Gibco, USA) and 1% penicillin-streptomycin (Gibco). The culture conditions were maintained in a humid incubator with temperature and $CO_2$ levels set at 37˚C and 5%, respectively.

### Preparation and analysis of HSC-MVs and TGF-β1HSC-MVs

HSC-MVs and TGF-β1HSC-MVs were generated from LX-2 cells treated with LX-2 and TGF-β1(10ng/ml), respectively. Upon reaching 80% confluence, LX-2 cells were subjected to serum-free medium for 72 h, and MVs were collected through differential centrifugation [13]. The morphology, size, and quantity of MVs were examined using transmission electron microscopy (TEM) and nanoparticle tracking analysis (NTA).

### Cellular model of vascular injury

HUVECs ($5×10^3$ cells/100 μL) were seeded in a 96-well culture plate and incubated until reaching 80% confluence. Subsequently, the cells were treated with $H_2O_2$ at concentrations ranging from 300 to 700 μM for 20 h. Cell morphology was observed under a microscope and cell viability was assessed using the CCK-8 assay (Cat # CK04; Dojindo, Japan). The IC50 value of $H_2O_2$ on HUVECs was determined to establish an in vitro model of endothelial cell injury [14].

### Cell proliferation detection

The HUVECs were cultured according to the aforementioned procedures. When cell confluence reached 80%, $2×10^8$/mL HSC-MVs and TGF-β1HSC-MVs were added. Meanwhile, 600μM $H_2O_2$ was introduced, and co-culturing continued for an additional 24h. Cell proliferation was assessed using the CCK-8 assay.

### Lactate dehydrogenase assay

Following the experimental procedure outlined for HUVECs, we measured lactate dehydrogenase (LDH) release using an LDH Activity Assay Kit (Dojindo, Japan, Catalog #: CK12), following the manufacturer's instructions.

## Cell apoptosis detection

The Annexin V-PE/7AAD Apoptosis Detection Kit (BD Biosciences, USA, Catalog #: 559763) was used to analyze apoptosis in treated HUVECs, following the manufacturer's instructions. After co-culture, HUVECs were fixed, stained with Annexin V-PE and 7-AAD solution, and subjected to flow cytometry using a BD FACSCalibur (USA).

## Reactive oxygen species (ROS) production analysis

Intracellular ROS levels were quantified using DCFH-DA (Cat #: CA1410; Solarbio, China), following the manufacturer's guidelines. After the aforementioned treatments, HUVECs were exposed to the DCFH-DA solution for 20 min at 37°C. The fluorescence intensity of ROS within the cells was observed using a fluorescence microscope. Five randomly selected fields were observed, and the cell count, and average were determined.

## Migration assays

Cell migration assays were conducted using transwell chambers (Falcon, Corning, USA, Cat#: 354234) equipped with a polycarbonate membrane. In the transwell migration assay, $7 \times 10^4$ HUVEC cells (200μL) were seeded in the upper chambers in 2.5% FBS medium. The upper chambers were treated with MVs (HSC-MVs or TGF-β1HSC-MVs) and 600 μM $H_2O_2$, while the lower chambers contained 20% FBS. After 16 h of incubation, cells in the upper chambers were removed, and cells in the lower chambers were stained with crystal violet (Beyotime, China, Cat#: C0121) at 25°C for 1 min. The cells were then observed and counted under a microscope. Cells from five randomly selected fields were counted and averaged.

## Tube formation assay

After thawing, the basement membrane matrix (Matrigel, Corning, USA, Cat#:354234) was dissolved at 4°C and added to 48-well plates at a volume of 100μl per well. The plates were then incubated at 37°C for 1 h. Next, $2 \times 10^4$ HUVECs were seeded in 200μl of serum-free endothelial cell medium and added to the Matrigel-coated wells. The control group was treated with medium alone, while the test group was treated with a mixture of MVs (HSC-MVs or TGF-β1HSC-MVs) and $H_2O_2$. Total tube branching length, segments, and junctions were calculated using the ImageJ software.

## Western blot

Proteins from cells were extracted using a lysis buffer. Protein lysates were separated using sodium dodecyl sulfate-polyacrylamide gel electrophoresis (SDS-PAGE) and transferred onto polyvinylidene fluoride (PVDF) membranes. The membranes were blocked for 1 h with 5% BSA and then incubated with primary antibodies against Cleaved caspase-3 (1:500, Cat#: ab2302), VEGF (1:1000, Cat#: ab32152), eNOS (1:1000, Cat#:ab199956), CXCR4 (1:1000, Cat#: ab181020), PI3K (1:2000, Cat#: ab140307), P-PI3K (1:1000, Cat#: ab182651), Akt (1:500, Cat#: ab8805), P-Akt (1:5000, Cat#: ab81283), Erk1/2 (1:10000, Cat#: ab184699), P-Erk1/2 (1:1000, Cat#: ab201015), GAPDH (1:5000, Cat#: ab8245), Tubulin (1:2000, Cat#: ab7291), and β-Actin (1:1000, Cat#: ab8226) (all from Abcam, USA) at 4°C overnight. After washing thrice for 30 min each with Tris-buffered saline Tween-20 (TBST), immunoreactivity was visualized using an ECL solution (Amersham, Sweden). Finally, grayscale maps of the bands were analyzed using ImageJ software.

## In vivo rat model of liver injury

Forty adult male Sprague-Dawley (SD) rats of SPF grade, weighing 180 and 220 g, were obtained from the Experimental Animal Center of Guangdong. The rats were acclimated in the Laboratory Animal Center of Guangdong Medical University for one week under controlled environmental conditions (23–25°C, 12-h dark/light cycle) before commencing the experiments. All the animals received appropriate care during the study with free access to food and water. This study received approval from the Animal Ethics Committee of Guangdong Medical University (NO. GDY1902010), adhering to the committee's guidelines for the care and use of laboratory animals.

A chronic liver injury model was established in rats by administering intraperitoneal injections of $CCl_4$ (50%, 1 ml/kg) twice weekly for 10 weeks [15]. The rats were randomly divided into four groups, each comprising 10 rats: (1) Control group: Rats received intraperitoneal injections of olive oil twice a week for 10 weeks; (2) $CCl_4$ group: Rats were intraperitoneally injected with $CCl_4$ twice a week for 10 weeks; (3) HSC-MVs-$CCl_4$ group: Rats received intravenous injections of stationary HSC-MVs at a concentration of 200μl ($2\times10^8$/ml) via the tail vein and intraperitoneal injections of $CCl_4$ twice a week for 10 weeks; (4) TGF-β1HSC-MVs-$CCl_4$ group: Rats were intravenously injected with TGF-β1HSC-MVs at a concentration of 200μl ($2\times10^8$/ml) via the tail vein and intraperitoneally injected with $CCl_4$ twice a week for 10 weeks.

Upon completion of the experiment, rats were subjected to cardiac blood collection using syringes under 2.5% isoflurane general anesthesia. Blood samples were sent to the Laboratory Department of Guangdong Medical University Affiliated Hospital for ALT and AST measurements. Under deep anesthesia, the rats were euthanized by cervical dislocation. Liver tissue was collected through an abdominal incision, with a portion fixed in 4% paraformaldehyde for 24 h, subsequently embedded in paraffin for measurement of histomorphology, immunohistochemistry and immunofluorescent staining. Another portion was rapidly frozen in liquid nitrogen and stored at -80°C for measurement of Western blot.

## Histological examination

The liver tissue samples were subjected to hematoxylin and eosin (H&E) staining to evaluate their morphological features. Masson's trichrome staining was performed to assess the degree of collagen damage in hepatic vessels.

## Immunohistochemistry

Paraffin-embedded liver tissues were sectioned into 4–6 mm pieces, followed by deparaffinization and sequential dehydration in ethanol. The sections underwent a 3-min boiling step and were immersed in EDTA-Tris (pH 9.0) for antigen retrieval. Subsequently, they were treated with hydrogen peroxide for 30 min to inactivate endogenous peroxidases. The sections were then subjected to an overnight incubation at 4°C with primary antibodies, including those against CD31 (1:2000, Abcam, USA, Cat#: ab182981), CD34 (1:200, Abclonal, China, Cat#: A0761), and αSMA (1:2500, Abcam, USA, Cat#: ab124964). Appropriate biotinylated secondary antibodies (goat anti-rabbit IgG and goat anti-mouse IgG; Abcam, Cambridge, USA) were added to the sections, which were then incubated for 30 min at room temperature(20~25°C). The sections were stained with diaminobenzidine (DAB) and counterstained with hematoxylin; the positive areas appeared brownish yellow. Immunohistochemistry results were analyzed using Image-Pro Plus software.

## Immunofluorescent staining

The samples were incubated with periodate-lysine-paraformaldehyde fixative at room temperature for 3 h. After cryoprotection with 30% sucrose/0.1M phosphate buffer (pH 7.2), approximately 20-μm thick cryostat sections were cut, and nonspecific staining was blocked by incubating the sections with 1% bovine serum albumin/PBS for 1 h. The sections were then incubated at room temperature for 24 h with primary antibodies, including Anti-CD31 (1:200, Abclonal, China, Cat# A2104), Anti-CD34 (1:200, Abclonal, China, Cat# A0761). Subsequently, the sections were washed three times with wash buffer (Beyotime) and incubated with a mixture of secondary antibodies at room temperature for 1 h. Negative control staining was performed by replacing the primary antibodies with control IgG for each immunized animal. The secondary antibodies used were Alexa Fluor 488 (Abcam, USA, Cat#: ab150077) conjugated donkey anti-rabbit and rat IgG, purchased from Abcam. Sections were observed under a confocal scanning laser microscope (Leica, TCS SP5II, Germany) after double labeling. For each animal, eight randomly chosen low-power optical fields were selected, and the number of positive cells was counted for each single channel and merged channel for each marker. Five animals were analyzed for each marker.

## Statistical analysis

Statistical analyses were performed using SPSS 25.0 software. Experimental results are expressed as mean ± standard deviation ($\bar{X}$ ± SD). Comparisons among multiple groups were conducted using one-way ANOVA, preceded by a homogeneity of variance test. Post-hoc comparisons were performed using either the LSD or Tamhane's test. Statistical significance was set at $P < 0.05$.

## Results

### $H_2O_2$ induced HUVECs injury model

$H_2O_2$ damaged HUVECs in a concentration-dependent manner. Fig 1A shows the morphological changes in HUVECs treated with different concentrations of $H_2O_2$. In the 0μM group, a single cell appeared as a short spindle or polygon with a plump morphology, and the confluent cell monolayer exhibited a cobblestone appearance. In the $H_2O_2$ group, cells displayed varying degrees of shrinkage, rounding, and shedding. $H_2O_2$ reduced HUVECs viability in a concentration-dependent manner. When HUVECs were exposed to varying concentrations of $H_2O_2$ (0, 300, 400, 500, 600, and 700μM), the cellular viability rate was negatively correlated with the concentration of $H_2O_2$ (r = 0.92, $P < 0.01$; Fig 1B). Using GraphPad software, the half maximal inhibitory concentration ($IC_{50}$) of $H_2O_2$ required to impair HUVECs was estimated to be 623.6μM (Fig 1C). Consequently, $H_2O_2$ at a concentration of 600μM was implemented in the subsequent experiments.

### Characteristics of MVs

Transmission electron microscopy (TEM) analysis revealed that the MVs had a round shape, intact membranes, and were present as single or multiple aggregates. The particle size ranged from 150 to 500 nm, consistent with the typical range of 100–1000 nm of MVs (Fig 1D and 1E).

The results of nanoparticle tracking analysis (NTA) indicated that HSC-MVs and TGF-β1HSC-MVs had sizes ranging from 100 to 600 nm, with concentrations around $2 \times 10^8$ /ml of cell culture medium (Fig 1F and 1G). These findings confirmed that the HSC-MVs and TGF-β1HSC-MVs isolated from the HSCs culture medium were indeed MVs.

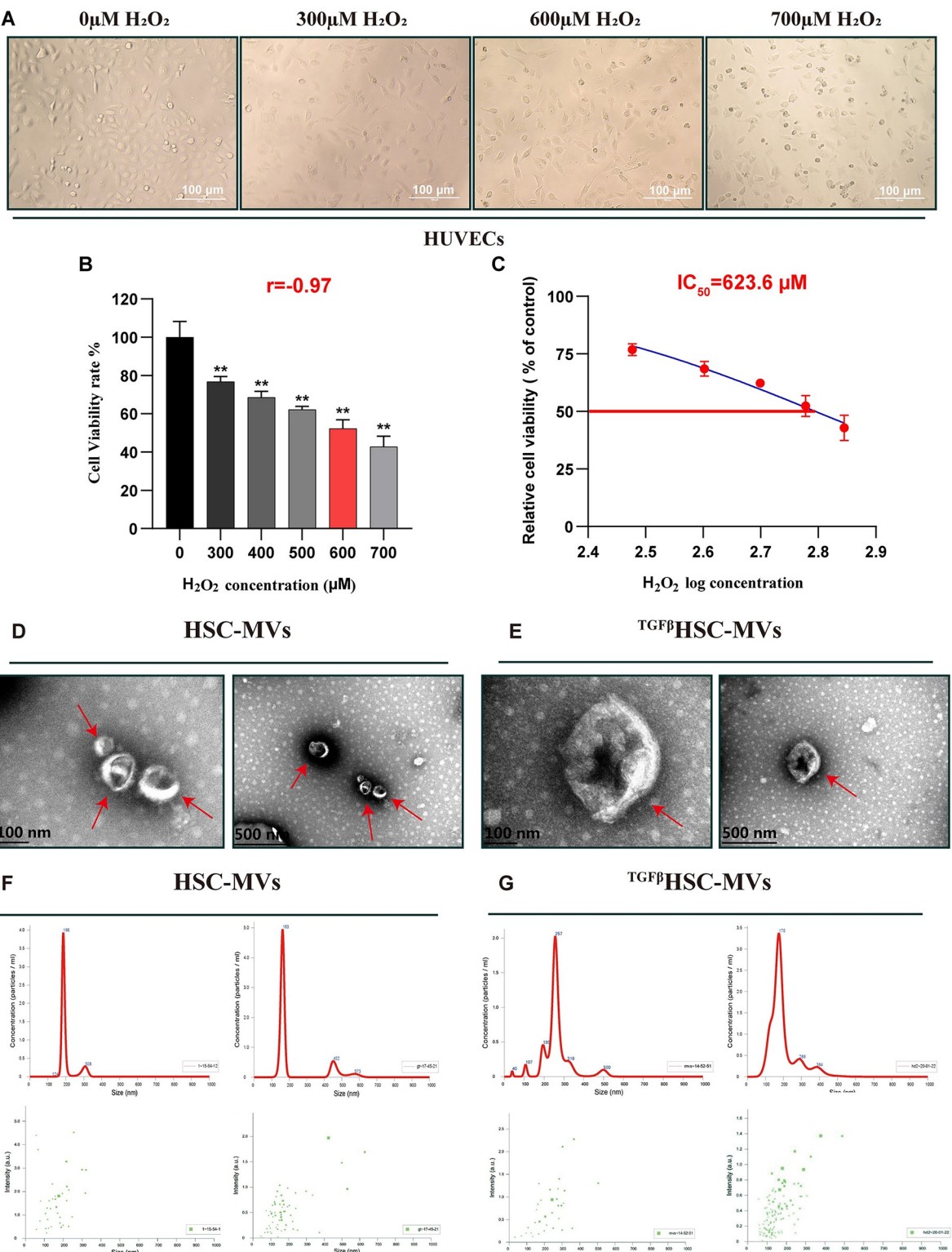

**Fig 1. The influence of different concentrations of H₂O₂ on the morphology and the viability rates of HUVECs and the characterization of HSC-MVs and TGF-β1HSC-MVs. A.** The morphology of HUVECs under an optical microscope, scale bar:100μm; **B.** The viability rates; **C.** $IC_{50}$ value of $H_2O_2$ on HUVECs viability rate. mean±SD, n = 6. *P<0.05, **P<0.01, vs 0μM group. $IC_{50}$: The half maximal inhibitory concentration, represents the concentration of $H_2O_2$ that is required for 50% inhibition of HUVECs. **D and E**. The size and morphology of HSC-MVs and TGF-β1HSC-MVs were detected using TEM, scale bar:100nm and 500nm; **F and G**. Particle size distribution in purified HSC-MVs and TGF-β1HSC-MVs were determined by NTA.

### HSC-MVs increased viability, reduced cytotoxicity, and inhibited apoptosis, while TGF-β1HSC-MVs decreased viability, enhanced cytotoxicity, and apoptosis in the $H_2O_2$ induced HUVECs injury model

Using the CCK8 proliferation assay, the results demonstrated that after 20 h of exposure to 600μM $H_2O_2$, compared with that of the control group, the HUVECs viability rate decreased by 37% (P<0.01). HSC-MVs significantly improved the proliferation ability of $H_2O_2$-injuried HUVECs; the viability rate increased by 13% compared with that of the $H_2O_2$ group (P<0.01). Conversely, TGF-β1HSC-MVs exacerbated the cellular injury effect of $H_2O_2$, with an 8% reduction in HUVECs viability rate compared with that of $H_2O_2$ group (P<0.01) (Fig 2A).

The lactate dehydrogenase (LDH) release assay and annexin V-PE/7-AAD analysis indicated that $H_2O_2$ exposure increased the cytotoxicity and apoptosis of HUVECs (vs. control group; P<0.01; Fig 2B and 2D). However, treatment with HSC-MVs significantly reduced the cytotoxicity and cell apoptosis rate (vs. the $H_2O_2$ group; P<0.01; Fig 2B and 2D). Conversely, treatment with TGF-β1HSC-MVs significantly increased the cytotoxicity and the cell apoptotic rate (vs. $H_2O_2$ group; P<0.01; Fig 2B and 2D). Additionally, western blot analysis showed that $H_2O_2$ promoted the expression of the apoptosis-related protein, Cleaved Caspase-3 (vs. control group; P<0.01). However, treatment with HSC-MVs inhibited the expression of Cleaved Caspase-3 compared with that in the $H_2O_2$ group (P<0.05). In contrast, treatment with TGF-β1HSC-MVs upregulated the protein expression of Cleaved Caspase-3 (vs. $H_2O_2$ group; P<0.01; Fig 2E and 2F).

### Differential effects of HSC-MVs and TGF-β1HSC-MVs on oxidative stress, cell migration and angiogenesis in $H_2O_2$-induced HUVECs injury

In the $H_2O_2$ induced HUVECs injury model, HSC-MVs improved oxidative stress response, cell migration ability and angiogenesis, whereas the effects of [TGF-β1]HSC-MVs were precisely the opposite.

$H_2O_2$ led to an elevation in ROS. DCFH-DA staining revealed that after 20h of exposure to 600μM $H_2O_2$, the number of ROS-positive cells in the $H_2O_2$ group increased by 4.4 times compared with that in the control group (p<0.01). Treatment with HSC-MVs significantly mitigated the elevated levels of ROS; the number of ROS-positive cells in the $H_2O_2$+HSC-MVs group decreased by 1.5 times compared with that in the $H_2O_2$ group (P<0.01). Conversely, treatment with TGF-β1HSC-MVs further increased the levels of ROS; the number of ROS-positive cells in the $H_2O_2$+TGF-β1HSC-MVs group increased by 1.2 times compared with that in the $H_2O_2$ group (P<0.01) (Fig 3A and 3D).

Transwell migration assay and crystal violet staining revealed that $H_2O_2$ inhibited the migration of HUVECs; the cell migration rate in the $H_2O_2$ group was 76% lower than that in control group (p<0.01). HSC-MVs effectively improved cell migration, while [TGF-β1]HSC-MVs further inhibited cell migration. Compared with that in the $H_2O_2$ group, the cell migration rate in the $H_2O_2$+ HSC-MVs group increased by 17%, while that in the $H_2O_2$+TGF-β1HSC-MVs group decreased by 14% (both P<0.01) (Fig 3B and 3E).

The in vitro angiogenesis experiment, also known as the endothelial tube formation assay, displayed morphological changes of cells incubated on the matrix gel for 24h (Fig 3C, 3F–3H). As shown in Fig 3C, the cells extend their protrusions, which come into contact with each other to form a capillary-like network structure. In the control group, the capillary-like network structure was clear, complete, and dense, while in the $H_2O_2$ group, the network structure was sparse and incomplete. In the $H_2O_2$+ HSC-MV group, the capillary-like network structure was similar to that in the control group. In the $H_2O_2$+TGF-β1HSC-MVs group, the capillary like network structure almost disappeared, with only fragments and scattered small branches

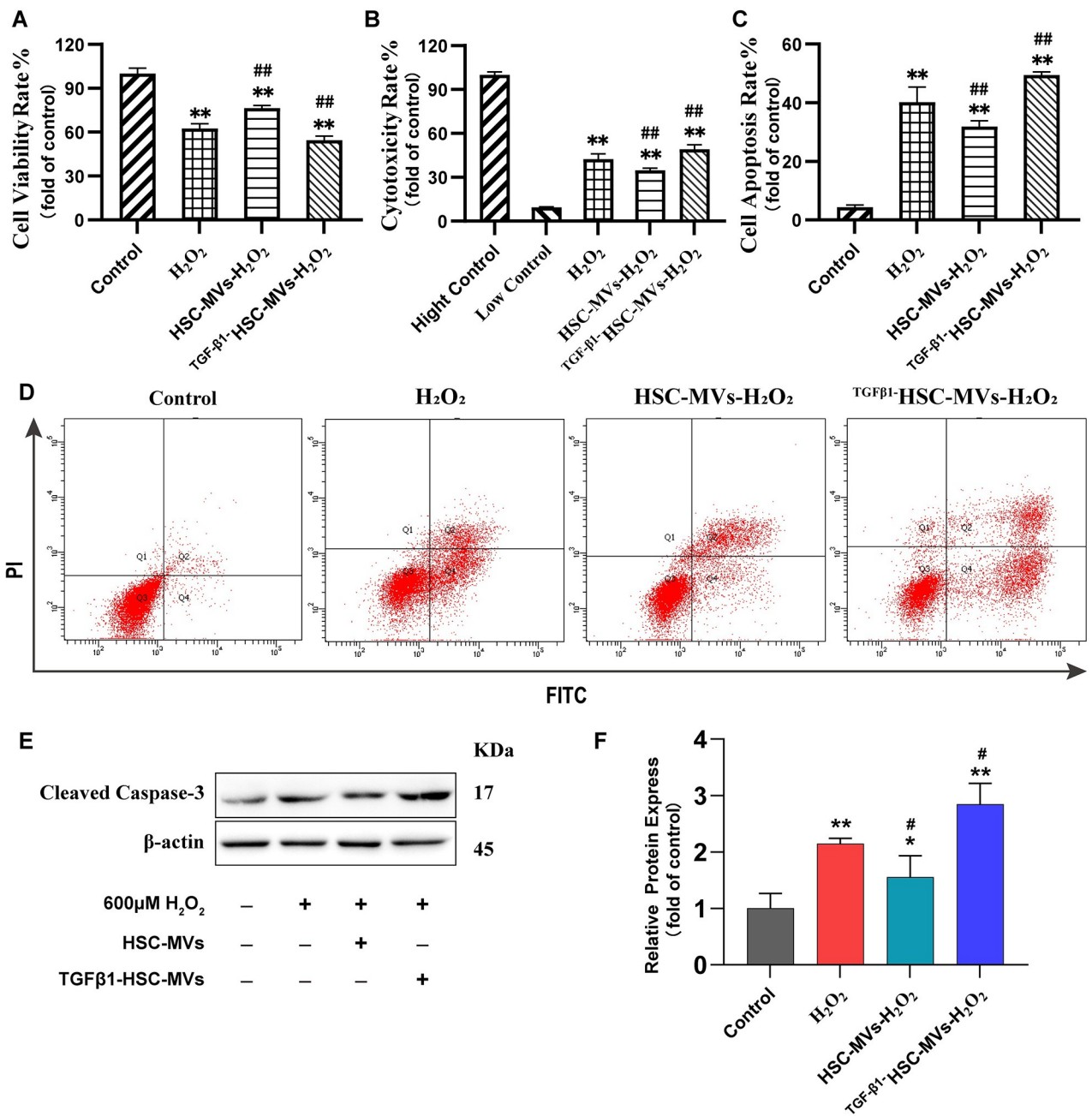

**Fig 2. Effects of HSC-MVs and TGF-β1HSC-MVs on H₂O₂ induced HUVECs injury. A.** Viability, (n = 4); **B.** Cytotoxicity, (n = 4); **C.** Statistical chart of apoptosis (n = 3); **D.** Flow cytometry graph of apoptosis; **E and F.** Cleaved Caspase-3 protein expression, (n = 3); mean ± SD. *p<0.05, **p<0.01, vs control group; #p<0.05, ##p<0.01, vs H₂O₂ group.

visible. The in vitro angiogenic capacity of each group was evaluated by analyzing the number of branch segments, branch intersections, and the total branch length in the capillary-like net-work structure. As shown in Fig 3F–3H, the number of branch segments (Nb Segments) (F), branch intersections (Nb Junctions) (G), and the total branch length (Tol. Branching Length) (H) in the H₂O₂ group were lower than that in the control group, with decreases of 48%, 41%, and 33%, respectively, all P<0.01. The H₂O₂+HSC-MVs group exhibited an increase of 21%,

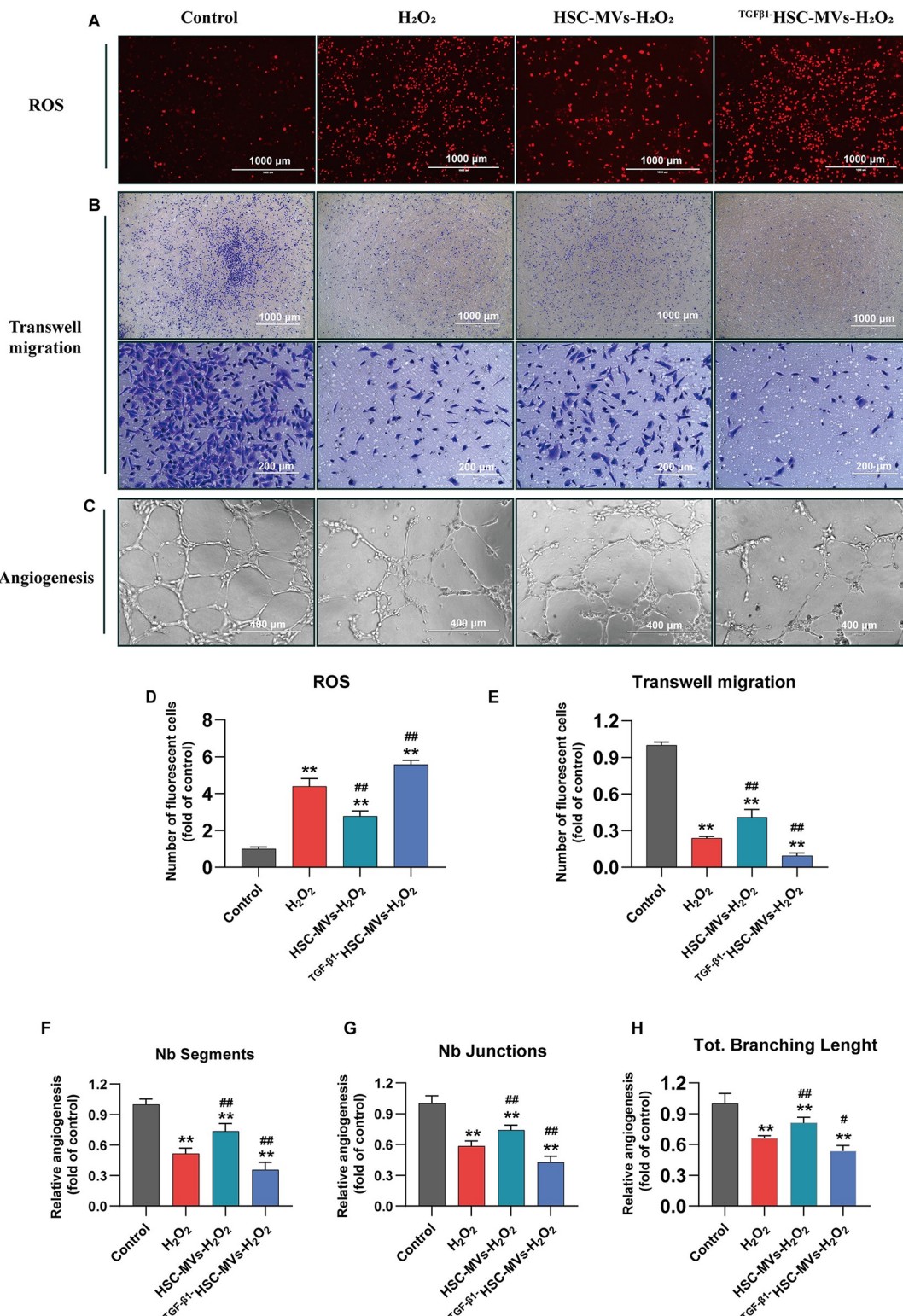

**Fig 3. The effects of HSC-MVs and TGF-β1HSC-MVs on oxidative stress, cell migration and angiogenesis in the $H_2O_2$ induced HUVECs injury model. A, D.** Reactive oxygen species (ROS), using DCFH-DA staining. **A**, Image under fluorescence microscope; **D**, Statistical Chart of Fluorescent Cell Count (n = 4). **B, E.** The cellular migration ability, using transwell chambers and crystal violet stained. **B**, Optical Microscopic Image of Crystal Violet Staining; **E**, Statistical chart of the number of transmembrane migrating cells (n = 4). **C, F, G, H** Angiogenesis. **C**, Image of angiogenesis under light microscope. F, G, H,

Statistical analysis of tube formation. The number of branch segments (Nb Segments) (F); the number of branch Junctions (Nb Junctions) (G), and the number of total branching length (Tol. Branching Length) (H), (n = 4). The data were expressed as mean ± SD. *p<0.05, **p<0.01, vs control group; #p<0.05, ##p<0.01, vs H₂O₂ group.

15%, and 14%, respectively, compared with the $H_2O_2$ group (all P<0.01). The $H_2O_2$+TGF-βHSC-MVs group showed a decrease of 15%, 15%, and 12% compared with the $H_2O_2$ group (all P<0.01) (Fig 3C and 3H).

## HSC-MVs upregulated the expression of PI3K, AKT, MEK1/2, ERK1/2, and angiogenesis-associated proteins in the $H_2O_2$-induced HUVECs injury model. Conversely, [TGF-β1]HSC-MVs inhibited the expression of these proteins

PI3K, AKT, MEK, ERK, etc. are important signaling molecules that regulate cell proliferation, differentiation, apoptosis, migration, and angiogenesis. The Western blot method was used to detect the protein expression of these signal molecules, and the results showed that no significant difference was noted in the total protein expression of PI3K or AKT between the control group and the $H_2O_2$ group, or $H_2O_2$+HSC-MVs group, or $H_2O_2$+TGF-β1HSC-MVs group. However, phosphorylated PI3K and AKT showed a downregulation of protein expression levels in $H_2O_2$ treated groups, all lower than the control group by 29% and 30%, p<0.01, compared with those in the control group. Static HSC-MVs upregulated the expression of phosphorylated PI3K or AKT proteins; the $H_2O_2$+HSC-MVs group was upregulated by 12% and 13% compared with the $H_2O_2$ group (p<0.01). TGF-β1 activated HSC-MVs further downregulated the expression of phosphorylated PI3K or AKT protein; $H_2O_2$+TGF-β1HSC-MVs group showed a 10% and 19% downregulation compared with the $H_2O_2$ group (p<0.05) (Fig 4A and 4B).

Similar to PI3K and AKT, there were no significant differences in the total protein expression levels of MEK1/2 or ERK1/2 between the control group and the $H_2O_2$ groups. However, $H_2O_2$ leads to the downregulation of phosphorylated MEK1/2 and ERK1/2 proteins. Compared with the control group, the $H_2O_2$ group downregulated by 27% and 30% (p<0.01). Static HSC-MVs can reverse the downregulation of phosphorylated MEK1/2 and ERK1/2 protein expression caused by $H_2O_2$, while [TGFβ1]HSC-MVs further exacerbate the downregulation of phosphorylated MEK1/2 and ERK1/2 protein expression. Compared with those in the $H_2O_2$ group, the protein expressions MEK1/2 and ERK1/2 were upregulated by 13% and 16% in the $H_2O_2$+HSC-MVs group (p<0.05); the $H_2O_2$+TGF-β1HSC-MVs group downregulated by 16% and 14% (p<0.05) (Fig 4C and 4D).

Signaling proteins related to angiogenesis, such as VEGF, eNOS, and CXCR4, react similarly to these proteins. $H_2O_2$ stimulation leads to the downregulation of VEGF, eNOS, and CXCR4 protein expression HSC-MVs can reverse the downregulation of protein expression, while TGF-β1HSC-MVs exacerbate the downregulation of protein expression. As shown in Fig 4E and 4F, the level of protein expressions of VEGF, eNOS, and CXCR4 in all groups treated with $H_2O_2$ were lower than those in the control group (all p<0.01). The expression of these three proteins in the $H_2O_2$+HSC-MVs group was higher than that in the $H_2O_2$ group (p<0.05). The expression of three proteins in the $H_2O_2$+[TGF-β1]HSC-MVs group was further downregulated, compared with that in the $H_2O_2$ group (p<0.05 or p<0.01).

## Effects of HSC-MVs and TGF-β1HSC-MVs on $CCl_4$-induced rat liver injury

Rats treated with $CCl_4$ exhibited a substantial increase in serum ALT and AST levels, surpassing those in the control group by 20 and 30 times, respectively (P<0.01). The combination of

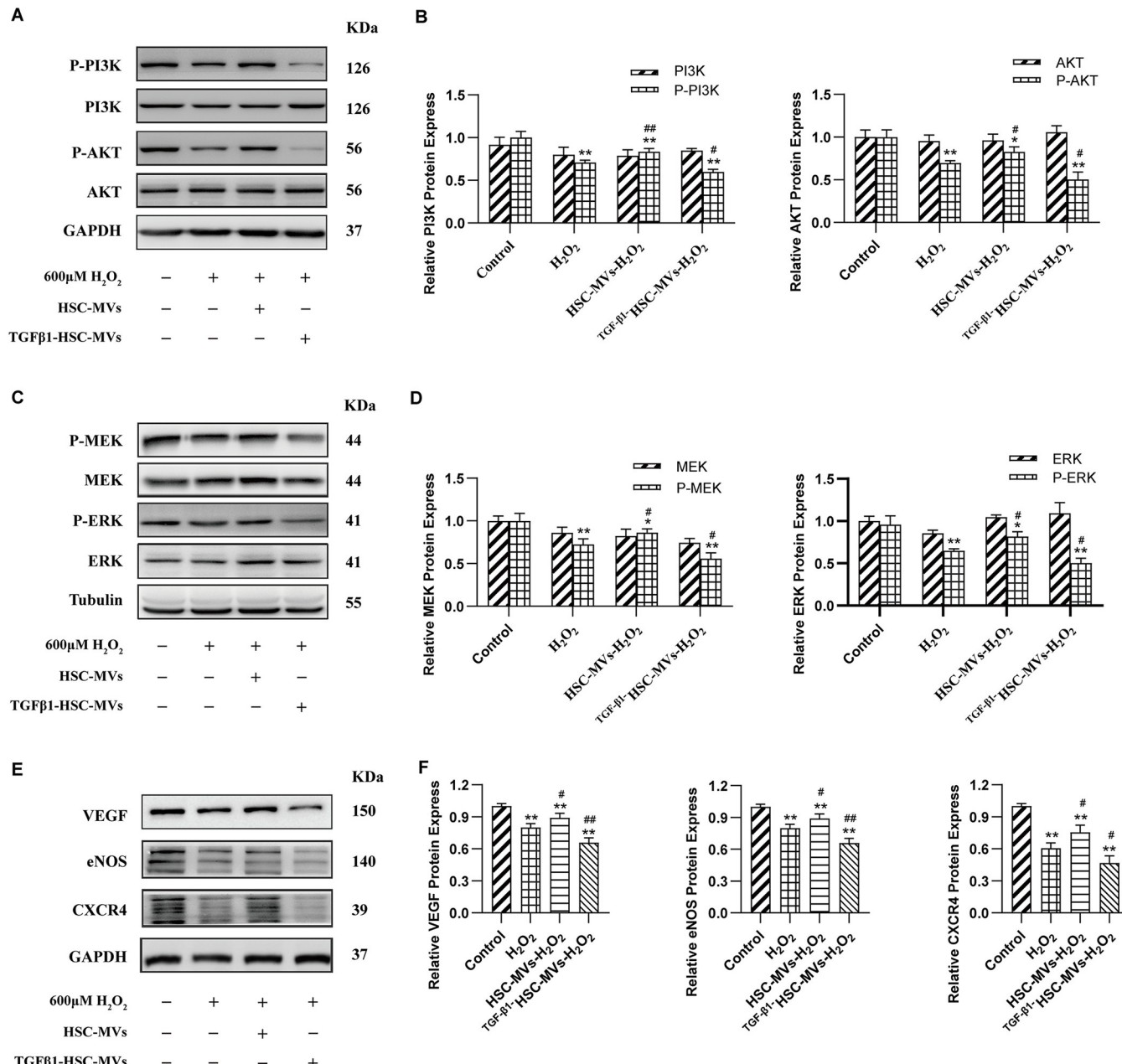

**Fig 4. The effects of static HSC-MVs and TGF-β1 stimulated HSC-MVs on the expression of signal proteins related to cell proliferation, apoptosis, oxidative stress, angiogenesis, etc. in the $H_2O_2$ induced HUVECs injury model. A to B**, $p$-PI3K, PI3K, $p$-AKT, AKT; **C to D**, $p$-MEK, MEK, $p$-ERT, ERT; **E to F**, VEGF, eNOS, CXCR4. A, C and E are the images displayed by western blotting; B, D and F are statistical charts. The expression levels were normalized with GAPDH or Tubulin. All experiments have been performed in triplicate and data were expressed as mean ± SD. *$p < 0.05$, **$p < 0.01$, vs. control group; #$p < 0.05$, ##$p < 0.01$, vs $H_2O_2$ group. $H_2O_2$ concentrations:600μM, HSC-MVs and TGF-β1HSC-MVs concentrations: $2 \times 10^8$ /ml.

HSC-MVs with $CCl_4$ resulted in a notable reduction in serum ALT and AST levels. Specifically, the HSC-MVs+$CCl_4$ group demonstrated a mere 1.1 and 1.5 times increase compared with the control group but exhibited a significant 30% and 40% decrease, respectively, in contrast to the $CCl_4$ -alone group (P<0.01). Conversely, co-administration of TGF-β1HSC-MVs with $CCl_4$ further increased serum ALT and AST levels. The TGF-β1HSC-MVs+$CCl_4$ group

displayed a 3-fold and 4.5-fold increase compared with the control group, and a 1-fold and 1.3-fold increase compared with the $CCl_4$- alone group (all P<0.01) (Fig 5A and 5B).

Histopathological examination of liver tissues revealed hepatocellular fatty degeneration and inflammatory cell infiltration. The combination of HSC-MVs and $CCl_4$ (HSC-MVs+$CCl_4$ group) mitigated the formation of liver fat vacuoles and the infiltration of inflammatory cells. In contrast, TGF-β1HSC-MVs (TGF-β1HSC-MVs+$CCl_4$ group) exacerbated the formation of hepatic fat vacuoles and the infiltration of inflammatory cells (Fig 5C). Masson's trichrome staining was used to assess the formation of fibrous tissue, fibrous septa, and perivascular collagen fibers in liver tissue. The $CCl_4$-alone group exhibited increased hyperplasia of fibrous tissue and the formation of fibrous septa and perivascular collagen fibers. Conversely, administration of HSC-MVs resulted in a significant decrease in the amount of fibrotic tissue and collagen fibers. However, treatment with TGF-β1HSC-MVs enhanced the levels of fibrosis tissue and collagen fibers (Fig 5D).

### Effects of HSC-MVs and TGF-β1HSC-MVs on CD31, CD34 and α-SMA expression in $CCl_4$-treated rats analyzed through immunohistochemistry staining and immunofluorescence staining

The results of immunohistochemistry and immunofluorescence showed minimal or negative expression of CD31 and CD34 in the control group, contrasting with a significant increase observed in the $CCl_4$ group. Co-treatment with HSC-MVs and $CCl_4$ led to a reduction in CD31 and CD34 expression, while TGF-β1HSC-MVs enhanced their expression. The integrated optical density (IOD) values, measured using Image-Pro Plus software, immunohistochemical results showed that the CD31 and CD34 IOD values in the $CCl_4$ group were 8 times and 6.5 times higher than those in the control group, respectively (both P<0.01). In the HSC-MVs+$CCl_4$ group, these values decreased by 57% and 52% compared with the $CCl_4$ group (both P<0.01); whereas in the TGF-β1HSC-MVs+$CCl_4$ group, they increased by 89% and 96% compared with those of the $CCl_4$ group (both P<0.01) (Fig 6A, 6B, 6E and 6F). Immunofluorescence results showed that the CD31 and CD34 IOD values in the $CCl_4$ group were 5-fold and 5.5-fold elevation, compared with the control group (both P<0.01). In the HSC-MVs+$CCl_4$ group decreased by approximately 48% and 56% compared with that in the $CCl_4$ group (both P<0.01). In contrast, the TGF-β1HSC-MVs+$CCl_4$ group exhibited an approximately 64% and 50% increase, compared with the CCl4 group (both P<0.01) (Fig 6C, 6D, 6G and 6H).

Furthermore, the impact of HSC-MVs and TGF-β1HSC-MVs on α-smooth muscle action (α-SMA) expression was examined through immunohistochemistry. Fig 7A and 7B illustrates that $CCl_4$ induced α-SMA expression, resulting in a significant increase compared with that in the control group. Co-treatment with HSC-MVs and $CCl_4$ attenuated α-SMA expression, while co-treatment with TGF-β1HSC-MVs and $CCl_4$ enhanced α-SMA expression. The IOD value measurement for α-SMA expression demonstrated that in the $CCl_4$ group, the IOD value was 5.3 times higher than that of the control group. The HSC+$CCl_4$ group exhibited a 48% decrease compared with the $CCl_4$ group, while the TGF-β1HSC-MVs+$CCl_4$ group increased by 1.3 times compared with the $CCl_4$ group (all P<0.01).

## Discussion

Extracellular vesicles (EV) consist of a heterogeneous population of nanosized particles enclosed in a lipid bilayer without a functional nucleus [16]. Depending on experimental conditions or the cellular release site (basolateral or apical face), the same cell type may secrete EVs with distinct cargo. EVs are classified into three discrete categories based on size and

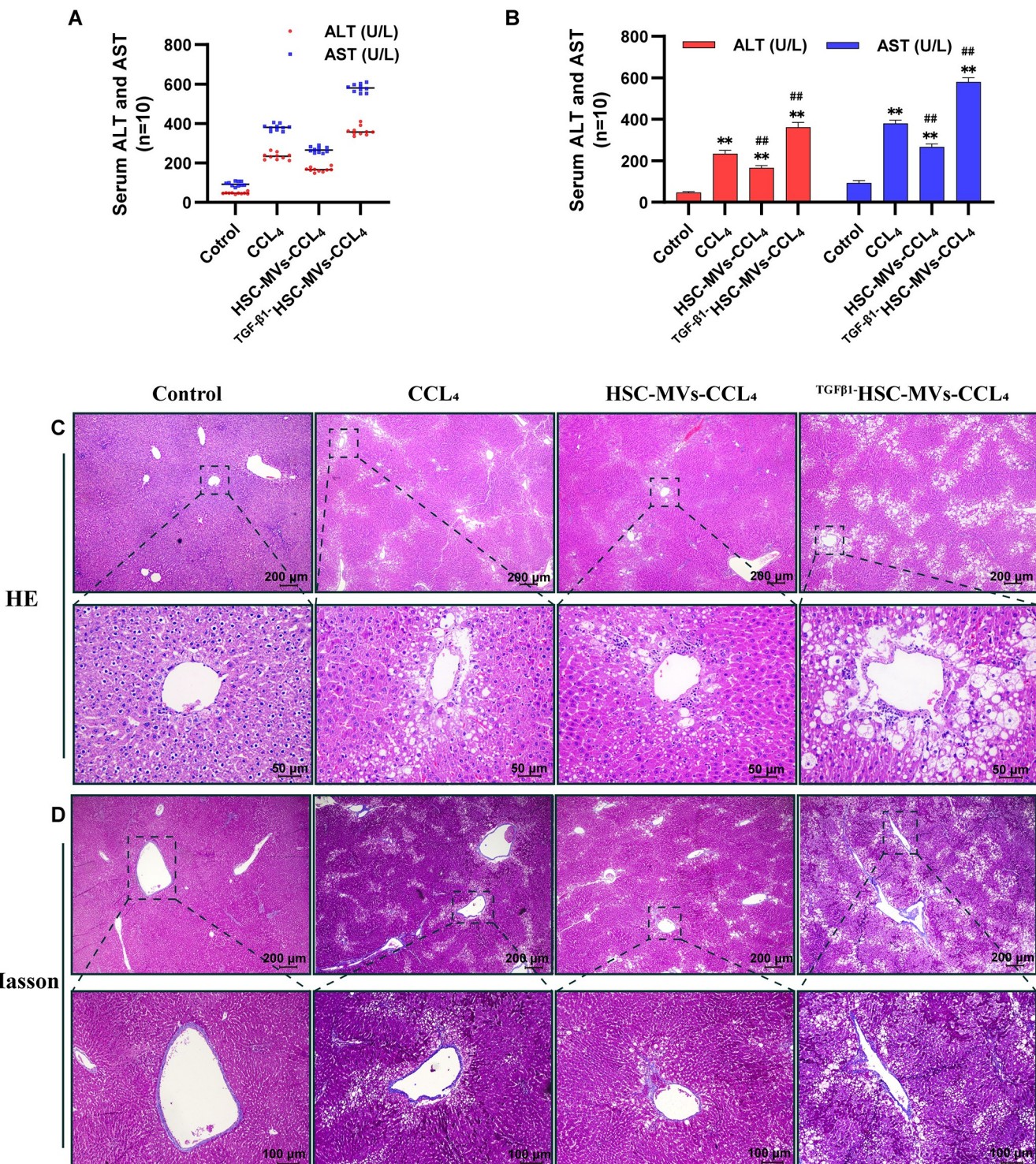

**Fig 5. The effects of static HSC-MVs and TGF-β1 stimulated HSC-MVs on the leakage of AST and ALT and the extent of liver fibrosis in CCl₄-treated rats. A B.** The levels of ALT and AST in rat serum, \*\*p<0.01, \*p<0.05 vs. control group; ##p<0.01, #p<0.05 vs. CCl₄ group (n = 10). **C.** H&E staining for liver sections, scale bars: 200μm and 50μm. **D.** Masson trichrome staining for liver sections, scale bars: 200μm and 100μm.

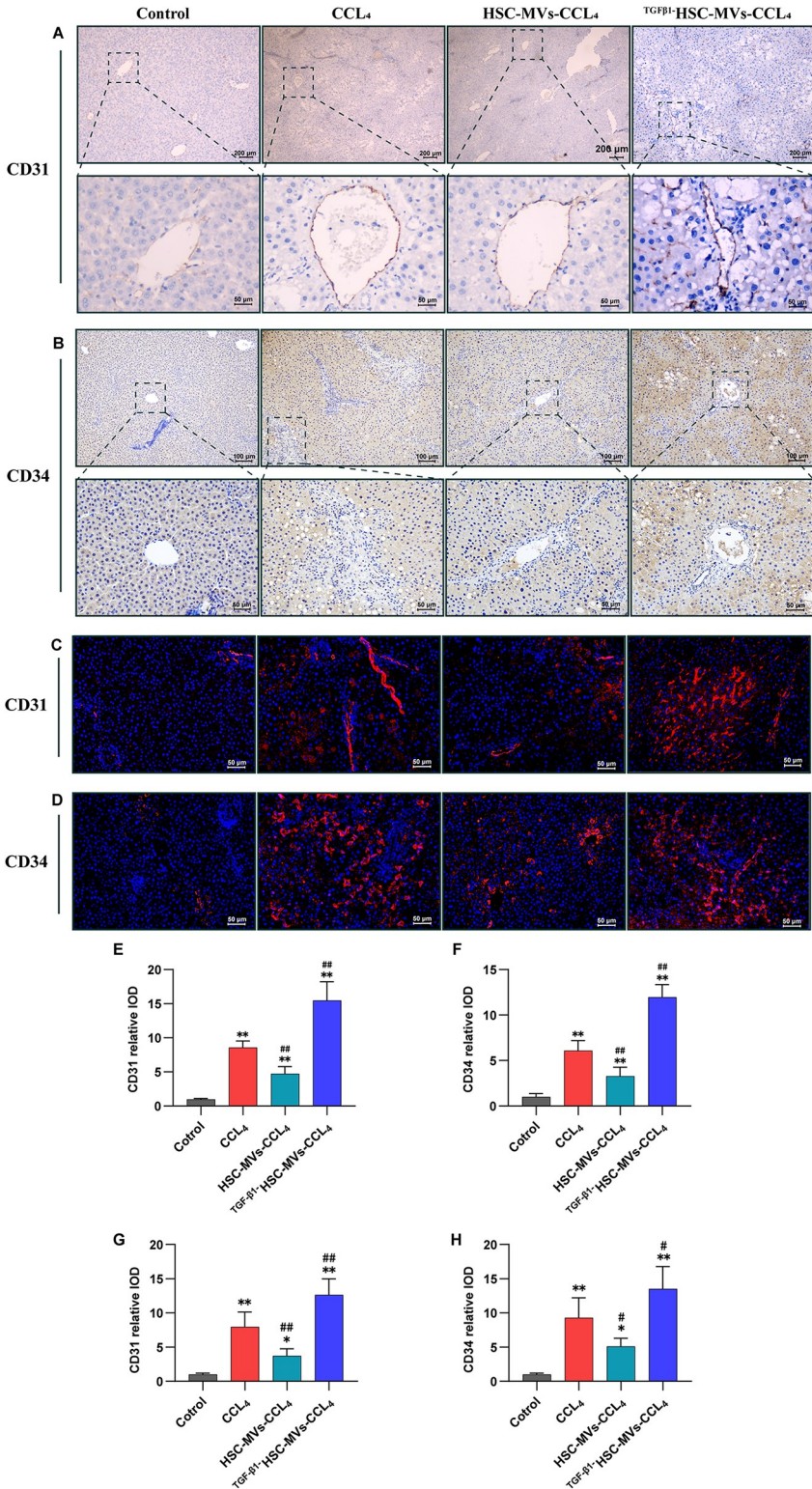

**Fig 6. The effects of static HSC-MVs and TGF-β1 stimulated HSC-MVs on the expression of CD31 and CD34 in CCl₄-treated rats using immunohistochemical and immunofluorescence staining method. A and B.** Immunohistochemical images. **A**. CD31, scale bars: 200μm and 100μm. **B**. CD34, scale bars:100μm and 50μm. **C and D.** Photo under a fluorescence microscope. Blue: Hoechst33342 staining, showed all cells. Red: Positive expression of proteins. **C.** CD31, scale bars: 50μm. **D.** CD34, scale bars: 50μm. **E to F**. Statistical graph of the integrated optical

density (IOD), IOD values were measured using Image-Pro Plus software. **p<0.01, *p<0.05 vs. control group. ##p<0.01, #p<0.05 vs. CCl₄ group. Each group consists of five rats, with three slices taken from each rat, and each slide took five pictures.

origin: exosomes, microvesicles, and apoptotic bodies [17]. Microvesicles, produced through outward budding and fission of the plasma membrane, are larger, ranging from 100 to 1000 nm [18]. In this study, nanoparticles were obtained using gradient centrifugation, with sizes ranging from 150 to 500nm (unstimulated HSC) or 100 to 600nm (TGF-β1 stimulated HSC). These particles exhibited a rounded shape and intact membranes, appearing as single or multiple aggregates under a transmission microscope. These results align with the reported surface characteristics of microvesicles in the literature [19].

To investigate the impact of HSC-MVs on vascular endothelial cell function, a cell model of HUVECs injury induced by $H_2O_2$ was employed. The results demonstrated concentration-dependent damage to HUVECs by $H_2O_2$, with concentrations ranging from 0 to 700μM. The cell survival rate exhibited a negative correlation with the concentration, (correlation

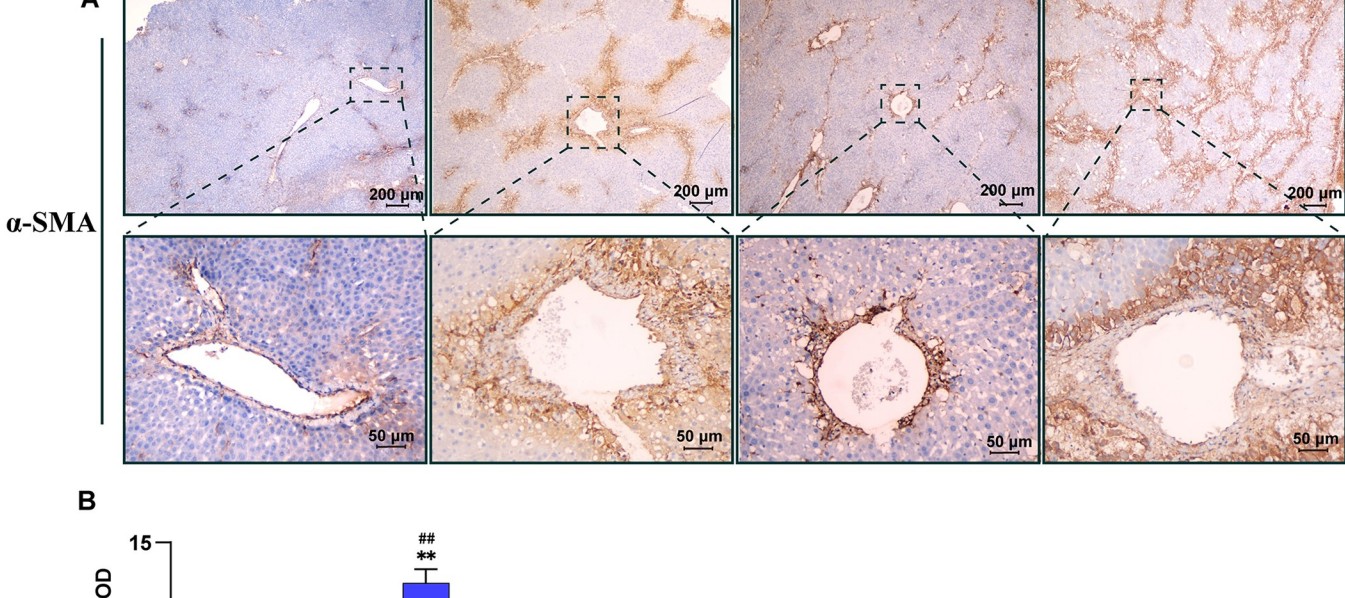

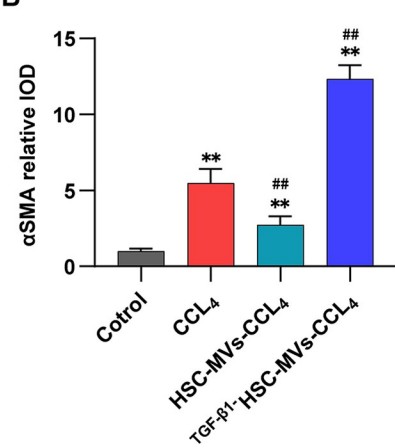

**Fig 7. The effects of static HSC-MVs and TGF-β1 stimulated HSC-MVs on the expression of α-SMA in CCl₄ induced rat liver injury using immunohistochemical method. A**. Immunohistochemical images of α-SMA, scale bars: 200μm and 50μm. **B.** Statistical graph of the integrated optical density (IOD), IOD values were measured using Image-Pro Plus software. **p<0.01, *p<0.05 vs. control group. ##p<0.01, #p<0.05 vs. CCl₄ group. Each group consists of five rats, with three slices taken from each rat, and each slide took five pictures.

coefficient, r = 0.92; half maximal inhibitory concentration, $IC_{50}$ = 623.6μM). After HUVECs were exposed to $H_2O_2$ for 20 h, the cell survival rate decreased by 45%, the apoptosis rate increased by 35%, and there was a 1.2-fold increase in the expression of cleaved caspase-3 protein. Reactive oxygen species (ROS) levels increased by 4.4 times, and lactate dehydrogenase release increased, while the cell migration rate and angiogenesis were inhibited. These results indicate that $H_2O_2$ detrimentally affects the morphology and functionality of HUVECs, confirming the successfully establishment of a damaged model.

Microvesicles represent a vital medium for intercellular communication [20]. Various bioactive substances, including proteins, lipids, carbohydrates, and nucleic acids, attach to the inner surface of the plasma membrane. During microvesicles formation, these substances are encapsulated, entering the intercellular space upon secretion, thereby exerting a biological influence on receptor cells [21]. The composition of bioactive substances released by microvesicles correlates with the microenvironment of the cells generating them [21]. The impact of HSC-MVs on vascular endothelial cell damage was explored in this study. The findings indicate that microvesicles released by hepatic stellate cells exhibit varied effects on $H_2O_2$ induced HUVECs injury under different conditions. MVs derived from unstimulated human LX-2 cells (human hepatic stellate cells) (HSC-MV) demonstrate cytoprotective properties, alleviating cell damage in $H_2O_2$-induced HUVECs injury. Concurrent exposure of HSC-MVs ($2 \times 10^8$/mL) and $H_2O_2$ (600μm) to HUVECs culture medium for 20 h resulted in increased cell survival rates, reduced cytotoxicity and apoptosis, improved oxidative stress response, enhanced cell migration ability, and increased angiogenesis. Conversely, microvesicles derived from TGF-β1-stimulated human LX-2 cells (TGF-β1HSC-MVs) exacerbated cell damage in $H_2O_2$-induced HUVECs injury. Simultaneous exposure of TGF-β1HSC-MVs ($2 \times 10^8$/mL) and $H_2O_2$ (600 μm) to HUVECs medium led to a further reduction in cell survival rates, increased cytotoxicity and apoptosis, aggravated oxidative stress, inhibited cell migration capacity and decreased angiogenesis.

The phosphatidylinositol 3-kinase (PI3Ks) protein family regulates various cellular functions such as proliferation, differentiation, and apoptosis [22]. AKT, also known as protein kinase B (PKB), is the primary downstream effector of PI3K. Activated AKT modulates cellular functions, such as anti-apoptosis and enhanced cell growth, by phosphorylating downstream factors, including enzymes, kinases, and transcription factors [23]. The MEK/ERK signaling pathway is involved in mediating diverse cellular functions, including proliferation, apoptosis, differentiation, migration and angiogenesis [24].

To determine whether the PI3K/AKT and MEK/ERK signaling pathways were responsible for the effects of HSC-MVs and TGF-β1HSC-MVs, we detected the expression of total proteins and phosphorylated proteins of PI3K, ARK, MEK1/2, and ERK1/2 in $H_2O_2$-induced HUVECs injury using western blot analysis. The results showed that $H_2O_2$ induced the downregulation of phosphorylated PI3K, AKT, MEKI/2, and ERK1/2. Static HSC-MVs mitigated the downregulation of phosphorylated PI3K, AKT, MEK1/2 and ERK1/2 protein expression caused by $H_2O_2$. In contrast, TGF-β1-activated HSC-MVs further exacerbated the downregulation of phosphorylated PI3K, AKT, MEK1/2 and ERK1/2 protein expression. These results affirm that HSC-MVs can activate the PI3K/AKT and MEK/ERK signaling pathways, while TGF-β1-activated HSC-MVs can inhibit the PI3K/AKT and MEK/ERK signaling pathways in $H_2O_2$-induced HUVECs injury.

Our data showed that $H_2O_2$ stimulation led to the downregulation of VEGF, eNOS, and CXCR4 protein expression. HSC-MVs reversed the effects of $H_2O_2$, upregulating VEGF, eNOS and CXCR4 protein expression, while TGF-β1HSC-MVs intensified the impact of $H_2O_2$, further downregulating VEGF, eNOS, and CXCR4 protein expression.

To validate the results of the in vitro experiments, we established an animal model of liver injury using $CCl_4$. The results showed an increase in serum ALT and AST, which were 20 and 30 times higher than the control group. Serum ALT and AST are important indicators for detecting whether liver function is normal [25]. Elevated serum ALT and AST levels indicate severe damage to liver function [26]. Histopathological examination of the liver revealed hepatocellular fatty degeneration, inflammatory cell infiltration, fibrous tissue hyperplasia, fibrous septa, and perivascular collagen fiber formation. In addition, the expression of vascular endothelial markers CD31 and CD34 increased. These results indicate liver vascular damage and compensatory proliferation of collagen fibers and the vascular endothelium.

The effects of different states on $CCl_4$-induced liver injury and hepatic vascular injuries were investigated in this study. Both intraperitoneal injection of $CCl_4$ and tail vein injection of unstimulated HSC-MVs ($2\times10^8$/mL) significantly improved liver and hepatic vascular injury. This was manifested by reduced serum ALT and AST levels, diminished liver tissue fat formation, decreased inflammatory cell infiltration, thinner collagen fiber bands, and downregulated CD31 and CD34 expression. These findings substantiate that unstimulated HSC-MVs have a protective effect against liver and hepatic vascular injury. However, simultaneous treatment with HSC-MVs ($2x10^8$/mL) stimulated by TGF-β1 and $CCl_4$ worsened liver injury and hepatic vascular injury. This was evidenced by increased serum ALT and AST levels, aggravated liver tissue adipogenesis, heightened inflammatory cell infiltration, thicker collagen fiber bands, and upregulated CD31 and CD34 expression, underscoring the aggravating effect of TGF-β1-stimulated HSC-MVs on liver injury and hepatic vascular injury. These results align with the in vitro experimental results.

This study used HUVECs and rats to investigate the effects of HSC-MVs on endothelial cells and vascular injury during liver damage. Preliminary findings demonstrated that static HSC-MVs have a protective effect against vascular injury, whereas TGF-β1-activated HSC-MVs exacerbate vascular injury. However, the study did not further analyze or investigate the related mechanisms and signaling pathways in vitro, resulting in some limitations. The protective and damaging effects of HSC-MVs on blood vessels still require further research. HSC-MVs may represent a new therapeutic target for liver injury and liver fibrosis.

## Conclusion

MVs produced by unstimulated HSCs (HSC-MVs) exert a protective effect on $H_2O_2$-induced HUVECs injury, whereas those produced by TGF-β1-stimulated HSCs (TGF-β1HSC-MVs) exacerbate $H_2O_2$-induced HUVECs injury. The effects of HSC-MV and TGF-β1HSC-MVs are mediated through the PI3K/AKT/VEGF, CXCR4, and MEK/ERK/eNOS signaling pathways. Simultaneously, HSC-MVs confer a protective effect against $CCl_4$-induced liver injury and hepatic vascular injury in rats, while [TGFβ1]HSC-MVs elicit an aggravating effect on $CCl_4$-induced liver injury and hepatic vascular injury in rats.

## Supporting information

**S1 Graphical abstract.**
(TIF)

**S1 Raw images.**
(PDF)

**S1 Data.**
(PDF)

## Acknowledgments

The authors extend their gratitude to Professors Yonghua Chen, Biyun Lin, and Yongfang Ou (Pathological Diagnosis and Research Center, Affiliated Hospital of Guangdong Medical University) for their technical assistance, as well as to Mr. Bingwen Miao, Jiahao Lei, Guoyun Wang, Junfeng Wang, Ms. Chujun Mo and Meijing Zhao (Department of Hepatobiliary Surgery, The Second Affiliated Hospital of Guangdong Medical University) for their invaluable help during the experiments.

## Author Contributions

**Conceptualization:** Jianlong Xie, Zhirong Ye, Qunwen Pan, Xiaotang Ma, Huilai Miao.

**Data curation:** Jianlong Xie, Zhirong Ye, Xiaobing Xu, Anzhi Chang.

**Formal analysis:** Jianlong Xie, Zhirong Ye, Xiaobing Xu, Anzhi Chang, Ziyi Yang, Qin Wu.

**Funding acquisition:** Huilai Miao.

**Investigation:** Jianlong Xie, Zhirong Ye, Anzhi Chang, Ziyi Yang, Qin Wu.

**Methodology:** Jianlong Xie, Xiaobing Xu, Yan Wang, Xiaotang Ma, Huilai Miao.

**Project administration:** Jianlong Xie, Xiaobing Xu, Yan Wang, Xiaotang Ma, Huilai Miao.

**Resources:** Xiaobing Xu, Qunwen Pan, Yan Wang, Yanyu Chen, Xiaotang Ma.

**Supervision:** Jianlong Xie, Xiaobing Xu, Yan Wang, Yanyu Chen.

**Validation:** Jianlong Xie, Xiaotang Ma, Huilai Miao.

**Visualization:** Jianlong Xie, Qin Wu.

**Writing – original draft:** Jianlong Xie, Anzhi Chang.

**Writing – review & editing:** Jianlong Xie, Xiaotang Ma, Huilai Miao.

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
