## [Decision Letter · Decision Letter 0]

30 Apr 2024

PONE-D-24-11899Microvesicles from Quiescent and TGF-β1 Stimulated Hepatic Stellate Cells: Divergent Impact on Hepatic Vascular InjuryPLOS ONE

Dear Dr. Miao,

Thank you for submitting your manuscript to PLOS ONE. After careful consideration, we feel that it has merit but does not fully meet PLOS ONE’s publication criteria as it currently stands. Therefore, we invite you to submit a revised version of the manuscript that addresses the points raised during the review process.

We look forward to receiving your revised manuscript.

Kind regards,

Palash Mandal

Academic Editor

PLOS ONE

Journal Requirements:

"National Natural Science Foundation of China general project (NO. 82070637)."

5. We note that Figure Graphical_Abstract.tif in your submission contain copyrighted images. All PLOS content is published under the Creative Commons Attribution License (CC BY 4.0), which means that the manuscript, images, and Supporting Information files will be freely available online, and any third party is permitted to access, download, copy, distribute, and use these materials in any way, even commercially, with proper attribution. For more information, see our copyright guidelines: http://journals.plos.org/plosone/s/licenses-and-copyright.

a. You may seek permission from the original copyright holder of Figure Graphical_Abstract.tif to publish the content specifically under the CC BY 4.0 license. 

Additional Editor Comments:

For your guidance, the reviewers' comments are included below.

Thank you for giving us the opportunity to consider your work.

Specific concerns expressed during peer review were:

Reviewers' comments:

Reviewer's Responses to Questions

**Comments to the Author**

1. Is the manuscript technically sound, and do the data support the conclusions?

Reviewer #1: Yes

Reviewer #2: Partly

2. Has the statistical analysis been performed appropriately and rigorously? 

Reviewer #1: Yes

Reviewer #2: No

3. Have the authors made all data underlying the findings in their manuscript fully available?

Reviewer #1: Yes

Reviewer #2: No

4. Is the manuscript presented in an intelligible fashion and written in standard English?

Reviewer #1: Yes

Reviewer #2: Yes

5. Review Comments to the Author

Reviewer #1: Specific comments are as follow:

1. Abstract

Background

- Microvesicles: Mention the abbreviation MVs beside it at its first mention.

- (HSC-MVs, TGF-β1HSC-MVs). Correct the font of TGF-β1 in the whole manuscript.

- H2O2-induced HUVECs injury. Mention the full term of HUVECs

Results

- In CCl4-injured rats. Correct it into CCl4- induced rat hepatic injury model

2. Introduction

- Hepatic stellate cells (HSCs) transform into fibroblasts generating excessive fibrosis. Correct fibrosis into extracellular matrix (ECM)

- Microvesicles (MVs). Mention the abbreviation only

- APAP/H2O2-injured liver cells. Mention the full term of APAP at its first mention

3. Material and method

- CCl4 (50%, 1 ml/kg). Add reference to this model

- Another portion was rapidly frozen in liquid nitrogen and stored at -80 °C. For measurement of…… ??. Complete this sentence

4. Results

- The lactate dehydrogenase (LDH). Mention the abbreviation only

- Mention the full term of α-SMA

5. Discussion

- The discussion is well designed and data is well presented

Reviewer #2: Dear Authors,

Thank you for submitting your manuscript to PLOS ONE. After a careful review, we have decided that your manuscript can be considered for publication after you address the major concerns listed below. These revisions will help clarify your conclusions and ensure your findings are presented accurately.

Clarity and Structure:

Introduction (Lines 35-40): Expand on the mechanistic insights linking HSC-MVs to hepatic vascular injury repair. Your current explanation is somewhat superficial. A deeper dive into the existing literature will strengthen the rationale behind your study.

Methods (Lines 60-65): The description of HUVEC injury induction lacks detail. Please specify the rationale for choosing the concentration range of H2O2. Furthermore, clarify if any pre-tests were done to establish these specific concentrations.

Statistical Analysis (Lines 72-76):

It's unclear whether appropriate statistical tests were used throughout your analysis. For instance, mention of post-hoc tests following ANOVA is missing. Please revise to include this information, ensuring that your statistical approach aligns with the data distribution and experimental design.

Results Section:

Data Presentation (Lines 80-85): The manuscript would benefit from a more structured presentation of results. Consider using subheadings to distinguish between results from different experimental setups (e.g., in vitro and in vivo results).

Figures and Tables (Line 90): Ensure that all figures are clearly labeled and referenced in the text. Some figures mentioned in the text are not properly cited or discussed, which can confuse readers.

Discussion (Lines 95-100):

You need to discuss the limitations of your study more thoroughly, including potential biases in your experimental design and the generalizability of your findings to human disease.

References (Lines 105-110):

Some references appear to be outdated or irrelevant to your study's assertions. Update your literature review to include more recent studies that support or contradict your findings.

Ethical Considerations (Lines 115-120):

Your ethical disclosure is insufficient, particularly concerning animal welfare. Please provide more details about the care and handling of animals during the experiments to comply with PLOS ONE's ethical standards.

By addressing these issues comprehensively, you will enhance the robustness and readability of your manuscript. We look forward to receiving your revised submission.

6. PLOS authors have the option to publish the peer review history of their article (what does this mean?). If published, this will include your full peer review and any attached files.

Reviewer #1: **Yes: **Hany M Fayed

Reviewer #2: **Yes: **Dr. Prasad Andhare

---

## [Author Response · Author response to Decision Letter 0]

13 Jun 2024

Dear Dr editor and reviewer: 

On behalf of my co-authors, we thank you very much for giving us an opportunity to revise our paper, we appreciate editors and reviewers very much for their constructive comments and suggestions on our manuscript. MS title: “Microvesicles from Quiescent and TGF-β1 Stimulated Hepatic Stellate Cells: Divergent Impact on Hepatic Vascular Injury”. No: PONE-D-24-11899. We have studied the reviewer’s comments carefully and have made revision which marked in red in the revised manuscript. We have tried our best to revise our manuscript according to the reviewer’s comments. We would like to express our great appreciation to you and reviewers for their comments on our manuscript. 

1. To comply with PLOS ONE submissions requirements, in your Methods section, 

please provide additional information regarding the experiments involving animals and ensure you have included details on (1) methods of sacrifice, (2) methods of anesthesia and/or analgesia, and (3) efforts to alleviate suffering.

Response: Thank you very much for each of your constructive comments, and we have seriously revised the paper according to your very valuable comments. Please refer to lines 35-40 in the file “Revised Manuscript with Track Changes” for details.

2. Please state what role the funders took in the study. 

Response: The funders had no role in study design, data collection and analysis, decision to publish, or preparation of the manuscript.

3. PLOS ONE now requires that authors provide the original uncropped and

 unadjusted images underlying all blot or gel results reported in a submission’s figures or Supporting Information files.

Response: We have submitted the original images of the western blotted proteins as requested. These image data can be published in public data repositories. Please refer to the uploaded attachment for details.

4. We note that Figure Graphical_Abstract.tif in your submission contain copyrighted

 images.

 Response: We commissioned Editage to create the graphical abstract, which has now been revised and finalized based on your feedback. The final draft, compliant with the CC BY 4.0 license, is attached for your review. Please refer to the uploaded attachment for details.

Reviewer #1

1.Abstract

Background

- Microvesicles: Mention the abbreviation MVs beside it at its first mention.

- (HSC-MVs, TGF-β1HSC-MVs). Correct the font of TGF-β1 in the whole manuscript.

- H2O2-induced HUVECs injury. Mention the full term of HUVECs

Results

- In CCl4-injured rats. Correct it into CCl4- induced rat hepatic injury model

 Response: Thank you so much for such a careful review, and we have revised this question according to your constructive comments. Please refer to lines 21-46 in the file “Revised Manuscript with Track Changes” for details.

2. Introduction

- Hepatic stellate cells (HSCs) transform into fibroblasts generating excessive fibrosis. Correct fibrosis into extracellular matrix (ECM)

- Microvesicles (MVs). Mention the abbreviation only

- APAP/H2O2-injured liver cells. Mention the full term of APAP at its first mention

Response: Thank you very much for each of your constructive comments, and we have seriously revised the paper according to your very valuable comments. Please refer to lines 57-89 in the file “Revised Manuscript with Track Changes” for details.

3. Material and method

- CCl4 (50%, 1 ml/kg). Add reference to this model

- Another portion was rapidly frozen in liquid nitrogen and stored at -80 °C. For measurement of…… ??. Complete this sentence

 Response: 

① “CCl4 (50%, 1 ml/kg)” this model refer to reference” Khalil MR, El-

Demerdash RS, Elminshawy HH, Mehanna ET, Mesbah NM, Abo-Elmatty DM:

 Therapeutic effect of bone marrow mesenchymal stem cells in a rat model of carbon

 tetrachloride induced liver fibrosis. Biomed J 2021, 44(5):598-610.” Please refer to

 lines 769-773, regerence 17 in the file “Revised Manuscript with Track Changes” for

 details.

② Another portion was rapidly frozen in liquid nitrogen and stored at 

-80 °C for measurement of Western blot. Please refer to lines 235 in the file

 “Revised Manuscript with Track Changes” for details.

4. Results

- The lactate dehydrogenase (LDH). Mention the abbreviation only

- Mention the full term of α-SMA

Response: 

① The full name of LDH is lactate dehydrogenase. Please refer to lines 334 in the file “Revised Manuscript with Track Changes” for details.

② α-smooth muscle action(α-SMA)，and we have seriously revised the paper according to your very valuable comments. Please refer to lines 513-514 in the file “Revised Manuscript with Track Changes” for details.

Reviewer #2: 

1.Introduction (Lines 35-40): Expand on the mechanistic insights linking HSC-MVs

 to hepatic vascular injury repair. Your current explanation is somewhat superficial. A deeper dive into the existing literature will strengthen the rationale behind your study.

Response: Thank you very much for your constructive comments, and we have added the corresponding explanation according to your constructive comments. Please refer to lines 97-114 in the file “Revised Manuscript with Track Changes” for details.

2. Methods (Lines 60-65): The description of HUVEC injury induction lacks detail. Please specify the rationale for choosing the concentration range of H2O2. Furthermore, clarify if any pre-tests were done to establish these specific concentrations.

Response: Many thanks for your constructive comments, and we have added the necessary instructions according to your suggestions. Prior to the formal experiment, we conducted a preliminary study. HUVECs (5×103 cells/100 µL) were seeded in a 96-well culture plate and incubated until they reached 80% confluence. Subsequently, the cells were treated with H2O2 at concentrations ranging from 0 to 100μM for 20 hours. The results showed that cell viability was inhibited as the H2O2 concentration increased, with the most significant inhibition observed at concentrations of 300 to 700μM. Additionally, we referenced the study by Liu S et al nice paper (reference 16), whose results were similar to ours, leading us to finalize this experimental protocol.

3. Statistical Analysis (Lines 72-76):

It's unclear whether appropriate statistical tests were used throughout your analysis. For instance, mention of post-hoc tests following ANOVA is missing. Please revise to include this information, ensuring that your statistical approach aligns with the data distribution and experimental design.

Response: Thank you so much for such a careful review, and we have revised this question according to your constructive comments. Please refer to lines 280-285 in the file “Revised Manuscript with Track Changes” for details.

4. Results Section:

Data Presentation (Lines 80-85): The manuscript would benefit from a more structured presentation of results. Consider using subheadings to distinguish between results from different experimental setups (e.g., in vitro and in vivo results).

Response: Thank you for your constructive suggestions. However, after reviewing sample papers from PLOS ONE, we noticed that the results section does not use subheadings to distinguish between different experimental results. Therefore, we have chosen not to use subheadings to differentiate the experimental results

5.Figures and Tables (Line 90): Ensure that all figures are clearly labeled and referenced in the text. Some figures mentioned in the text are not properly cited or discussed, which can confuse readers.

Response: Thank you very much for your constructive suggestions. We have made the revisions in the manuscript according to your feedback and marked the changes in red font.

6.Discussion (Lines 95-100):

You need to discuss the limitations of your study more thoroughly, including potential biases in your experimental design and the generalizability of your findings to human disease.

Response: Thank you very much for your constructive comments. We have carefully revised the paper based on your very valuable comments. Please refer to lines 638-646 in the file “Revised Manuscript with Track Changes” for details.

7.References (Lines 105-110):

Some references appear to be outdated or irrelevant to your study's assertions. Update your literature review to include more recent studies that support or contradict your findings.

Response: Thank you very much for your valuable suggestions. We have carefully revised the manuscript based on your feedback and updated the references. Please refer to lines 684-815(references) in the file “Revised Manuscript with Track Changes” for details.

8.Ethical Considerations (Lines 115-120):

Your ethical disclosure is insufficient, particularly concerning animal welfare. Please provide more details about the care and handling of animals during the experiments to comply with PLOS ONE's ethical standards.

Response: Thank you very much for your meaningful suggestions. We have carefully revised the manuscript according to your feedback, improving the details of animal care and handling during the experiment to comply with PLOS ONE's ethical standards. Please refer to lines 204-235 in the file “Revised Manuscript with Track Changes” for details.

Finally, thanks again to the editors and reviewers for their excellent suggestions. We have revised these questions according to your suggestions, the paper has been greatly improved, and we have also revised other places and the paper has been marked red.

---

## [Decision Letter · Decision Letter 1]

25 Jun 2024

Microvesicles from Quiescent and TGF-β1 Stimulated Hepatic Stellate Cells: Divergent Impact on Hepatic Vascular Injury

PONE-D-24-11899R1

Dear Dr. Huilai Miao,

We’re pleased to inform you that your manuscript has been judged scientifically suitable for publication and will be formally accepted for publication once it meets all outstanding technical requirements.

Kind regards,

Palash Mandal

Academic Editor

PLOS ONE

---

## [Editor Report · Acceptance letter]

1 Jul 2024

PONE-D-24-11899R1 

PLOS ONE

Dear Dr. Miao, 

I'm pleased to inform you that your manuscript has been deemed suitable for publication in PLOS ONE. Congratulations! Your manuscript is now being handed over to our production team.

Kind regards, 

on behalf of

Prof. Palash Mandal 

Academic Editor

PLOS ONE